# Superiority of Cellulose Non-Solvent Chemical Modification over Solvent-Involving Treatment: Application in Polymer Composite (part II)

**DOI:** 10.3390/ma13132901

**Published:** 2020-06-28

**Authors:** Stefan Cichosz, Anna Masek

**Affiliations:** Lodz University of Technology, Institute of Polymer and Dye Technology, Faculty of Chemistry, Stefanowskiego 12/16, 90-924 Lodz, Poland; stefan.cichosz@dokt.p.lodz.pl

**Keywords:** cellulose, silane, green chemistry, polymer composite, non-solvent treatment

## Abstract

The following article debates on the properties of cellulose-filled ethylene-norbornene copolymer (EN) composites. Natural fibers employed in this study have been modified via two different approaches: solvent-involving (S) and newly developed non-solvent (NS). The second type of the treatment is fully eco-friendly and was carried out in the planetary mill without incorporation of any additional, waste-generating substances. Composite samples have been investigated with the use of spectroscopic methods (FT-IR), differential scanning calorimetry (DSC), static mechanical analysis, and surface-free energy measurements. It has been proved that the possible filler-polymer matrix interaction changes may occur due to the performed modifications. The highest reinforcement was evidenced for the composite sample filled with cellulose treated via a NS approach—TS = (34 ± 2) MPa, Eb = (380 ± 20)%. Additionally, a surface free energy polar part exhibited a significant increase for the same type of modification. Consequently, this could indicate easier wetting of the material which may contribute to the degradation process enhancement. Successfully developed cellulose-filled ethylene-norbornene copolymer composite compromises the rules of green chemistry and sustainable development by taking an advantage of renewable natural resources. This bio-inspired material may become an eco-friendly alternative for commonly used polymer blends.

## 1. Introduction

Nowadays, various fillers are employed in polymer industry in order to control the properties of a composite material, e.g., mechanical reinforcing [1,2,3], increasing thermal resistance [4,5], adjusting processing properties [6,7], creation of electrically conductive [8,9] or flame-retardant [10,11] polymer blends. As the final product is a system composed of many components, different substances offer various opportunities for altering its characteristics.

Interestingly, in recent years, a new filler group, which may revolutionize plastic industry, has risen. It is created by substances derived from the natural environment, renewable feedstock or biomass, e.g., cellulose [12,13,14], lignocellulose [15,16], silicates [17,18]. They not only may influence the mechanical, thermal, or processing properties of a polymer composite, but also they could contribute to the creation of a new generation of materials which are less harmful to the environment [19,20,21,22,23]. This could be achieved by exchanging some synthetic, non-biodegradable polymer composite components with their natural analogues [24,25]. 

An important group of bio-fillers is represented by natural fibers [26,27,28,29]. They may exhibit different properties depending on their origin and extraction method [30]. Moreover, what should be emphasized, while mixing the synthetic polymer matrix with cellulose-based fibers, one could be able to obtain a material of a higher degradation potential [31,32]. 

On the other hand, when the polymer matrix itself is biodegradable, e.g, thermoplastic starch [33], poly(lactic acid) [34,35], a fully biodegradable polymer composite could be obtained. Unfortunately, biopolymers cannot be employed in every application like their synthetic analogues [36]. Therefore, modification of a commonly used polymer matrix with a bio-filler seems to be an interesting option. Below some properties of cellulose-filled systems are presented [37,38,39].

Shumigin et al. [37] compared the influence of untreated cellulose fibers on poly(lactic acid) (PLA) and low density polyethylene (LDPE). Authors detected the higher melt viscosity, especially at low angular frequencies (then cellulose phase contribution becomes apparent). Moreover, cellulose-filled polymer composite zero shear viscosity was higher in comparison with the neat polymer matrix—for PLA increase by 16%, in case of LDPE—raise by 76%. Moreover, the following trend has been detected—the higher the volume fraction and droplet size of cellulose phase, the higher is the G’ at low angular frequencies. Therefore, G’ may be altered with the amount of natural fibers incorporated into the system. Furthermore, with increasing angular frequency, the filler effect decreases and the matrix contributions dominate. Considering the performance of the analyzed materials in static conditions, the tensile strength of cellulose-filled PLA oscillated around 55 MPa, but a filler caused a decrease in elongation at break from 9.3% to 6.5%. Similarly, regarding LDPE, cellulose incorporation caused a decrease in both tensile strength (from 18 MPa to 14 MPa) and elongation at break (from 78–54%).

This research reveals that the surface modification of plant-based fibers is essential [40,41]. Otherwise, the adhesion between the polar filler and hydrophobic polymer matrix is poor [42]. As a consequence, not sufficient mechanical and thermal performance of a composite material is observed. 

Ifuku et al. [38] proposed the modification of cellulose fibers with aminopropyltriethoxysilane (APS) in order to improve their adhesion to propylene glycol diacrylate (PGDA). The treatment has been carried out in ethanol/water (80/20) solution with different concentrations of APS. Authors managed to improve the composite sample tensile strength from 33.7 MPa (non-modified cellulose) to 42.7 MPa. At the same time, the material became stiffer—Young’s modulus increased from 826 MPa to 1506 MPa and elongation at break decreased from 12.4% to 8.7%. Moreover, fracture SEM images revealed that untreated fibers were pulled out from the matrix, while the fracture was smooth in case of treated cellulose-filled PGDA. Fibers were supposed to be broken off at the fractured surface and thus do not slipped through interface. Detected mechanical performance improvement was explained by the enhancement in the interfacial adhesion caused by the silane coupling agent.

On the other hand, Qian et al. [39] created a fully biodegradable polymer composite based on the poly(lactic acid) (PLA) and bamboo residues (BR) modified with the (3-mercaptopropyl)trimethoxysilane (MPTMS). BR have been soaked in NaOH solution, bleached with NaClO_4_ solution, and hydrolyzed with sulfuric acid prior to the silanization process (water/methanol mixture, pH = 4). Tensile strength of modified cellulose-filled polymer composites decreased from 20 MPa to 14 MPa. Yet, the elongation at break value elevated—from 150% to almost 400%. Authors observed a relatively long plastic deformation process after a rapid elastic deformation of the material. Fracture SEM images indicated that pure PLA exhibited typical brittle rupture without wire-drawing appearance, while untreated fibre-filled PLA—a short wire-drawing appearance. Treated fibre-filled PLA exhibited rather long wire-drawing appearance and a rugged surface (PLA stretched in the direction of tension). This phenomenon is a perfect example how filler-polymer matrix interactions may change upon the silanization process of cellulose.

Therefore, this kind of cellulose fibers modification is being the subject of many scientific researches [43,44,45,46,47]. The main aim of these works is to develop as good thermal and mechanical properties of polymer composites as possible. In order to achieve this goal, silane coupling agents seem to be very promising regarding their ability to hydrophobized the surface of cellulosic materials [48].

The following article debates on the properties of ethylene-norbornene copolymer composites filled with silane-treated cellulose fibers. The modification process has been carried out via two different approaches, only one of which involved solvent employment. It should be emphasized that solvent presence in the modification process, undoubtedly, contributes to the increased toxicity, hazard, pollution, and rise of waste [49]. This states against the rules of green chemistry and challenges the idea behind the creation of an eco-friendly alternative to commonly used polymeric materials. The process of eco-friendly polymer composite production should be non-toxic as well. Therefore, the presented research was carried out in order to answer for the first time the question if the non-solvent modification process may be efficient enough to create a cellulose-filled polymer composite of a sufficient performance.

## 2. Materials and Methods

### 2.1. Materials

The Arbocel^®^ UFC100 Ultrafine Cellulose for Paper and Board Coating (UFC100) from J. Rettenmaier & Soehne (Rosenberg, Germany) was employed in the following research study. Its density is approximately 1.3 g/cm^3^. Cellulose is insoluble in most of the commonly used polar/non-polar solvents. On the other hand, it is of a high water binding capacity (even at high temperatures and shearing forces). Average fibre length varies between 6 and 12 μm.

Cellulose fibers were treated with three types of silane: (i) triethoxyoctylsilane (TEOS) from Sigma-Aldrich (Poznan, Poland); (ii) trimethoxypropylsilane (TMPS); (iii) vinyltrimethoxysilane (VTMS) U-611 from UniSil (Tarnow, Poland). The chemical compositions of the silanes employed in this research are revealed in Figure 1.

Ethanol (concentration: 96%) was employed in the role of reaction media and was bought from Chempur (Piekary Śląskie, Poland). Ethanol exhibits the viscosity of 1.078 mPas (20 °C), and the density approximately 0.79 g/cm^3^ (20 °C). The vapor pressure at 20 °C is about 233 mbar. The solvent is soluble in, e.g., water.

Ethylene-norbornene copolymer (EN), TOPAS^®^ Elastomer E-140 from TOPAS Advanced Polymers^®^ (Raunheim, Germany) was employed in the role of polymer matrix. Figure 2 reveals the structure of the discussed copolymer. This material is a high-performance thermoplastic elastomer becoming an interesting alternative to traditional flexible materials for use in, e.g., healthcare appliances, injection molded articles (mostly optical industry), food packaging. The material exhibits: the melting temperature of 84 °C, the Vicat softening temperature of 64 °C, the bulk density from 450–550 g/dm^3^. 

### 2.2. Modification of Cellulose Fibres

The gathered information regarding the carried out modifications and sample name abbreviations are listed in Table 1.

#### 2.2.1. Solvent-Involving Modification

Ethanol was a reaction environment in this type o treatment (ethanol [ml] to cellulose [g] ratio—20:1). Cellulose and silane (cellulose [g] to silane [ml] ratio—3:1) were stirred in rotary evaporator in the presence of NH_4_
^.^ H_2_O (cellulose [g] to NH_4_
^.^ H_2_O [ml] ratio—15:2), in a flask for 2 h at 60 r/min (oil bath, 40 °C). Next, the vacuum distillation (oil bath 60 °C, initial pressure 200 mbar) has been performed. While the solvent has been removed, the samples were dried in the laboratory dryer for 4 h at 120 °C and, then, for 24 h at 100 °C. Prepared specimens were stored in the laboratory oven at 40 °C. The scheme of the silanization process has been revealed in Figure 3.

#### 2.2.2. Non-Solvent Treatment

This modification approach is introduced in this research. It has been performed with the employment of the planetary mill (Pulverisette 5, Fritsch, Merazet, Poznan, Poland). Cellulose and silane (cellulose [g] to silane [ml] ratio—3:1) were placed in two steel containers (10 steel milling balls of 5 mm diameter in each vessel). Modification conditions: 2 h and 300 rpm. Similarly to the solvent-involving approach, specimens were dried in the laboratory oven for 4 h at 120 °C and next for 24 h at 100 °C. Samples were stored in the dryer at 40 °C.

### 2.3. Preparation of Polymer Composite Samples

Before incorporation into the polymer matrix, cellulose fibers were dried for 24 h at 100 °C (Binder® oven, Tuttlingen, Germany; crystallizer 70 × 40 mm). Then, both components, namely, polymer matrix (93 wt%) and cellulose (7 wt%) have been put into the micromixer (Brabender Lab-Station from Plasti-Corder (Duisburg, Germany) with Julabo (Seelbach, Germany) cooling system) at 110 °C for 30 min (50 rpm). Prepared mixture was plasticized at 100 °C for 30 min in the laboratory oven and placed between two rolling mills (cellulose fibers orientation within the material) with 100 × 200 mm rolls; rolls temperature of 20–25 °C, friction of 1:1.1, 60 seconds. Finally, the composite plates were compressed between two steel molds, between two Teflon sheets, in a hydraulic press (electrically heated platens)—160 °C, 10 min, 125 bar. 

### 2.4. Characterization of Cellulose-Filled Polymer Composites

#### 2.4.1. Fourier-Transform Infrared Spectroscopy (FT-IR)

Fourier transform infrared spectroscopy (FT-IR) absorbance spectra has been investigated within the 4000–400 cm^−1^ range (64 scans, resolution of 4 cm^−1^, absorption mode). The experiment has been performed with the use of Thermo Scientific Nicolet 6700 FT-IR spectrometer equipped with diamond Smart Orbit ATR sampling accessory.

#### 2.4.2. Static Mechanical Analysis (SMA)

Tensile strength (TS) and elongation at break (Eb) were evaluated with the use of Zwick-Roel Z005 measuring device (Zwick-Roel, Wroclaw, Poland). Tests were performed with “dumbbell” shape samples (1.5 mm thick and 4 mm width) according to PN-ISO 37:1998 standard (punch described in a standard). Moreover, in order to observe the fibre orientation effect, samples have been cut out in two directions: vertically (marked as “|”) and horizontally (marked as “-“).

#### 2.4.3. Differential Scanning Calorimetry (DSC)

Differential scanning calorimetry (DSC) investigation was carried out in the following temperature range: −40–200 °C (heating rate: 10 °C/min; argon atmosphere). During the experiment, glass transition temperatures of ethylene elastic segments (Tg_1_) and rigid norbornene segments (Tg_2_) were determined, as well as material softening enthalpy (ΔH). Mettler Toledo TGA/DSC 1 STARe System equipped with Gas Controller GC10 has been employed (Mettler Toledo, Greifensee, Switzerland).

#### 2.4.4. Surface Free Energy (SFE)

Three liquids have been employed during the experiment: distilled water, ethylene glycol, 1,4-diiodomethane. Moreover, surface of polymer composites has been cleaned with acetone before the contact angle measurements (liquid droplets of 1 μL). In the surface tension experiment, OCA 15EC goniometer by DataPhysics Instruments GmbH (Filderstadt, Germany) equipped with single direct dosing system (0.01–1 mL B. Braun syringe, Hassen, Germany) was employed. The values of total surface free energy (E), as well as polar (*E_p_*) and dispersive part (*E_D_*) of the surface free energy were determined with the employment of the Owens–Wendt–Rabel–Kaelble (OWRK) method [51]:(1)$$E=E_{P}+E_{D}$$
(2)$$\frac{\sigma_{L}\left(1+\cos\Theta \right)}{2\sqrt{\sigma_{L}^{D}}}=\sqrt{\sigma_{S}^{P}}\cdot \sqrt{\frac{\sigma_{L}^{P}}{\sigma_{L}^{D}}}+\sqrt{\sigma_{S}^{D}}\;as\;a\;linear\;function:Y=a\cdot X+b$$
(3)$$while:\;Y=\frac{\sigma_{L}\left(1+\cos\Theta \right)}{2\sqrt{\sigma_{L}^{D}}},\;X=\sqrt{\frac{\sigma_{L}^{P}}{\sigma_{L}^{D}}},\;a=\sqrt{\sigma_{S}^{P}},\;b=\sqrt{\sigma_{S}^{D}}$$
(4)$$therefore\;E_{P}=a^{2}=\sigma_{S}^{P}\;and\;E_{D}=b^{2}=\sigma_{S}^{D}$$
where:
E—total surface free energy [mJ/m^2^]$E_{P}$—polar part of surface free energy [mJ/m^2^]$E_{D}$—dispersive part of surface free energy [mJ/m^2^]$\sigma_{L}$—total liquid surface tension [mN/m]$\sigma_{L}^{P},\sigma_{L}^{D}$—respectively: polar and dispersive part of liquid surface tension [mN/m]$\sigma_{S}^{P},\;\sigma_{S}^{D}$—respectively: polar and dispersive part of solid surface tension [mN/m]Θ—contact angle [◦]


### 2.5. Characterization of Cellulose Fibres

Further description of the performed modification effect on the properties of cellulose fibers has been discussed in the *Part I* of this research [50]. Nevertheless, some data considering the modified fibers characteristics are also considered in this article. The methods are given below.

#### Dynamic Light Scattering (DLS)

The hydrodynamic radius of cellulose particles dispersed in the distilled water solution (0.1 g of the powder per 200 mL of distilled water) was established with the employment of dynamic light scattering technique. Before the experiment, discussed solutions were subjected to ultrasound (30 min). Next, dispersion samples were placed in colorimetric cuvettes and the measurement was done. The used device: ZetaSizer Nano–S90 from MalvernInstruments (Malvern, UK). 

## 3. Results and Discussion

### 3.1. Fourier-Transform Infrared Spectra Investigation

Fourier-transform infrared spectroscopy have been carried out in order to investigate the chemical structure of polymer composite samples filled with modified and untreated cellulose fibers, as well as to analyze the possible changes in the FT-IR spectra.

In Figure 4 a typical spectra of neat ethylene-norbornene copolymer and unmodified cellulose-filled polymer composite are revealed. Some absorption bands characteristic of the polyolefin may be detected: C–H stretching (2915 cm^−1^, 2848 cm^−1^) [52,53], C–H bending vibrations (1463 cm^−1^) [54], CH_2_ rocking vibration (719 cm^−1^) [54]. Full list of absorption bands assigned to the appropriate chemical groups visible in the cellulose-filled ethylene norbornene copolymer composite structure is presented in Table 2.

As it could be observed in Figure 4, analyzed material may have become slightly oxidized during the processing. This might be evidenced by some spreaded signals in the region 1300–1100 cm^−1^ (C–O, C=O, C=C, –C–O–C–) [56] and 3600–3200 cm^−1^ (–OH) [53,58]. Nevertheless, while interpreting the FT-IR spectra of ethylene-norbornene copolymer given in Figure 4, one should remember that this is not a pure polyolefin. It is modified with norbornene rings incorporated in the form of the blocks between the ethylene segments. Therefore, some new signals due to the additional norbornene content might occur and slightly influence the shape of FT-IR spectra.

Furthermore, regarding Figure 4, some changes in the following regions become visible due to the cellulose loading (EN + UFC100): 720 cm^−1^ (C–C) [54], 1050 cm^−1^ (C–O) [54], 1300–1100 cm^−1^ (C–O, C=O, C=C, –C–O–C–) [56] and 3600–3200 cm^−1^ (–OH) [53,58]. The variations in these regions are the effect of some additional carbon- and oxygen-rich moieties presence in cellulose fibers. Nevertheless, most of the changes are visible between 2000 and 400 cm^−1^. This is why the further analysis of the composite specimens is carried out regarding only this region.

Figure 5 and Figure 6 reveal the chemical structure of composite samples filled with cellulose modified via, respectively, solvent-involving and non-solvent approach. Analyzing the FT-IR spectra of the mentioned specimens, it may be said that no significant differences in the chemical structure of a polymer composite filled with differently treated bio-filler may be evidenced.

Yet, some changes occur. The most important ones are visible considering the peaks in the following regions: 1700–500 cm^−1^ (C=C, C=O) [53,57] and 1200–1000 cm^−1^ (C–O, C=O, C=C, COOH) [56]. These are typical absorption bands of natural fibers.

Moreover, considering the composite sample filled with VTMS modified cellulose via a solvent-involving approach, an additional peak at 1743 cm^−1^ (C=O) [57] is visible. It could originate from the oxidized C=C bond. Moreover, only in the structure of VTMS the C=C bond exists. However, this absorption band is visible only for the solvent-involving modification. This indicates that solvent presence in the reaction media may have an impact on possible oxidation of C=C bond in the VTMS structure.

Summarizing, all cellulose-filled ethylene-norbornene copolymer composites exhibit a similar chemical structure and no significant differences between the FT-IR spectra are detected. Yet, some shifts between the same absorption bands visible in Figure 5 and Figure 6, e.g., 561 cm^−1^ to 562 cm^−1^, 1047 cm^−1^ to 1058 cm^−1^, 1597 cm^−1^ to 1596 cm^−1^, may indicate some information about the change of interactions between the modified fibers and polymer matrix. Consequently, this could lead to some differences in the mechanical and thermal properties of analyzed composite specimens.

### 3.2. Static Mechanical Analysis

Performed static mechanical tests have revealed differences between the investigated polymer composite samples concerning their performance, e.g., elongation at break, tensile strength, Young’s moduli.

Figure 7 reveals the changes in the moduli values regarding specimen elongations by 100%, 200% and 300%. It is clearly visible that polymer composites filled with cellulose fibers modified via a non-solvent approach exhibit higher moduli values and, therefore, the material becomes stiffer [39]. On the other hand, polymer composite samples filled with cellulose modified via a solvent-involving approach are not that stiff. Young’s moduli values are slightly lower in comparison with their analogues prepared with a non-solvent method. 

Observed phenomenon may be explained by the significantly decreased size of cellulose fibers after a mechano-chemical treatment [59,60]. Smaller particles may contribute to the material stiffening and its reinforcement [61].

Interestingly, also orientation of the filler plays an important role regarding the mechanical properties of the prepared composite samples as cellulose fibers are high-aspect ratio particles [42]. It is visible that some specimens are achieving higher elongation values in only one direction of cutting out of the samples, e.g., EN + UFC100/VTMS/S, EN + UFC100/TMPS/NS. Nevertheless, some specimens filled with, e.g., VTMS or TMPS treated fibers are unable to elongate by 300%—they are destroyed before achieving this elongation value.

This evidences, once more, the significance of filler distribution within a polymer matrix and the influence of the modifying agent. Appropriate filler alignment and proper treatment may be crucial considering achieving a composite sample of an increased performance [62].

Furthermore, Figure 8 reveals the tensile strength and elongation at break values for analyzed polymer composite samples. In most cases the lowering of composite mechanical performance may be observed. The reinforcing effect of the modified cellulose only slightly differs from untreated natural fibers.

Nevertheless, reproducing the tensile strength of the neat polymer matrix is possible. Regarding Figure 8a, cellulose modified with TMPS via a non-solvent approach, while incorporated into the ethylene-norbornene copolymer, significantly reinforce the material creating a product of a similar performance to the neat polymer matrix. On the other hand, the elongation at break of this material is lower in comparison with pure EN. This is the effect of material stiffening caused by the hydrophobized fibers incorporation to the system and, simultaneously, evidence the filler-polymer matrix interaction improvement [1,42].

Comparing the obtained results, it may be claimed that non-solvent and solvent-involving approaches of cellulose modification give similar results regarding the filler behavior within the polymer matrix. Yet, in some cases the performance of the obtained material may be higher owing to the mechano-chemically modified cellulose incorporation. 

This could be explained by the fact that during such a treatment, not only the cellulose fibers grafting with silane coupling agent occurs, but also the particle size decreases due to the milling process [59,60]. The combination of an efficient hydrophobization of natural fibers and appropriate particle size distribution may be the way to obtain a cellulose-reinforced polymer composite. Nevertheless, changes in the filler dispersion and composite morphology should be taken into consideration in order to fully understand the ongoing variations in mechanical properties.

### 3.3. Differential Scanning Calorimetry Analysis

Differential scanning calorimetry (DSC) was used in order to establish the effect of the filler modification approach on the glass transition temperature of both ethylene (Tg_1_) and norbornene (Tg_2_) segments. Furthermore, during the measurement it was possible to observe the process of material softening—its enthalpy was determined (ΔH). 

In Figure 9 DSC curves of composites filled with neat and modified cellulose fibers are presented. It is visible that some variations between the analyzed samples occur regarding their thermal behavior. Regarding data given in Figure 9b–d it may be concluded that silanization of natural fibers via two different approaches has a similar effect on the ethylene-norbornene copolymer composite properties.

Some more details revealing the slight differences between the DSC curves presented in Figure 9 are given in Table 3. It is visible that the cellulose incorporation slightly influence glass transition temperature regimes and the softening enthalpy values.

Furthermore, on the basis of data gathered in Table 3 it may be claimed that the type of cellulose modification influences the glass transition temperature of elastic ethylene segments in a varied way and has a slight impact on the rigid norbornene rings behavior. 

Generally, Tg_1_ values could be considered as slightly higher for cellulose fibers modified via a mechano-chemical approach. This may indicate some information about increased interactions between the hydrophobized cellulose fibers and a polymer matrix [63]. This could also explain the previously observed material stiffening and shifts in the FT-IR spectra. 

Considering the values of enthalpy change assigned to the softening process of the material, it is visible that all composite samples filled with the modified cellulose fibers exhibit similar values. Nevertheless, they are higher in comparison with the neat UFC100 filled specimen. Similar behavior has been detected in different studies [64,65].

This, again, could indicate some information about the filler-polymer matrix interactions development [63]. What should be emphasized, there is no impact of cellulose incorporation, whether it is treated or not, on the peak temperature assigned to the softening process. This effect was also reported in different research studies [66,67].

Summarizing, regarding the DSC data, there is no significant difference in the thermal behavior of analyzed polymer composite samples. Nevertheless, some filler-polymer matrix interaction improvement possibility, being a reason for some slight changes in glass transition temperatures and softening enthalpy, has been detected.

### 3.4. Surface Free Energy Analysis

Among different modifications of cellulose fibers that were carried out, some changes in composite sample surface free energies were noticed. As a consequence, taking into consideration the gathered results, not only the cellulose surface energy is altered during its modification [68,69], but also the composite sample surface energy is altered by the fibers incorporation [70].

Graph describing the surface free energy changes is presented in Figure 10. What could be seen is the lower energy for all cellulose-filled composite samples in comparison with the neat ethylene-norbornene copolymer. Furthermore, modified filler incorporation leads to the increase in the polar part of surface free energy.

Furthermore, regarding the differences between the two modification approaches, it may be concluded that, in general, surface free energy exhibit similar level throughout all of performed modifications. The only exception is the EN + UFC100/TMPS/NS—here, an elevated E value could be detected. 

Moreover, it could be claimed that non-solvent modified cellulose fibers incorporation into EN results in the higher possibility of the surface free energy polar part increase. This is significant considering the ecological issues as the wetting properties depend not only on total surface free energy, but they also vary considering the polar/dispersive components [71]. Therefore, water would be able to wet easier the surface which possess a higher value of surface energy polar part, as H_2_O is a highly polar solvent. This, in turn, may contribute to the easier and quicker degradation of the cellulose-filled material [72].

Gathering all of the gathered data, cellulose fibers lead to the fall of polymer composite surface free energy regarding both solvent-involving and non-solvent modification method. Yet, non-solvent approach enables more intense increase in surface free energy polar part which may help in developing new opportunities for the creation of eco-friendly and sustainable polymer composites.

### 3.5. Influence of Cellulose Characteristics on the Properties of Polymer Composite Sample

Taking into consideration the results from the cellulose structure analysis which is a subject of the previous part of this research study, it could be concluded that the fibre characteristics have a great influence on the filled polymer composite properties. In Figure 11 some possible trends are presented.

Regarding Figure 11a, it could be observed that the TMPS possessing 3 carbon atoms in its alkyl chain is the most appropriate silane for the cellulose modification. This kind of treatment enables obtaining the best possible mechanical properties of a polymer composite. On the other hand, VTMS could have had too short alkyl chain in order to provide the efficient cellulose hydrophobization and octyl chain in TEOS structure might have resulted in too strong interfacial filler-polymer matrix interactions, simultaneously, causing the material stiffening [73].

Moreover, Figure 11b indicates that the size of the fibre, and not the silane coupling agent structure, has a crucial influence on the Tg of elastic ethylene segments in the ethylene-norbornene copolymer. Elevated values are observed for the fibers which exhibit lower hydrodynamic radii.

On the basis of gathered information it is visible that both chemical grafting of cellulose and fibre size have an influence on the polymer composite behavior. Mechano-chemical treatment provides an opportunity to adjust the fibre length and modify the cellulose surface at the same time which makes this method uniform.

### 3.6. Comparison of Prepared Composite Properties with LDPE

Low-density polyethylene (LDPE) is a commonly used material in food packaging [74,75]. This is the branch of the industry where newly developed cellulose-filled polymer composites being the subject of this article could be employed.

In Table 4 the comparison between these two materials is presented. It may be observed, that EN filled with cellulose fibers modified with TMPS via a non-solvent approach (sample exhibiting the best mechanical performance and increased polar part of surface free energy) reveals higher tensile strength than pure LDPE and has elongation at break in the region of the LDPE matrix which is very promising for the future.

As developed polymer composite exhibits relatively better properties (Table 4) than the commonly employed in food packaging low-density polyethylene (LDPE), bio-based ethylene-norbornene copolymer composite, in future, may become an interesting alternative to the materials used in the packaging industry.

## 4. Conclusions

Comparing the data presented in this article (Table 5), it may be concluded that more significant changes upon cellulose polymer composite properties occur while the non-solvent cellulose modification process is employed in order to hydrophobize the bio-filler. The treatment method is solvent-free and eco-friendly. Not only does it provide a chemically altered product, but also enables the fibre size decrease which is crucial considering the properties of polymer composite materials.

Furthermore, some of the investigated composite samples exhibited an improved tensile strength and elevated polar part of the surface free energy. Water would wet easier the surface that possess a higher value of surface energy polar part and contribute to the easier degradation of the cellulose-filled material. The highest differences concerning different parameters were evidenced for the EN with addition of UFC100/TMPS/NS.

What is important, developed cellulose-filled ethylene-norbornene copolymer composite compromises the rules of green chemistry and sustainable development by taking an advantage of renewable natural resources. Therefore, analyzed material may have a possibility of easier degradation because of the incorporation of natural fraction, which makes it promising, considering the food packaging applications. Nevertheless, further analysis is required.

## Figures and Tables

**Figure 1 materials-13-02901-f001:**
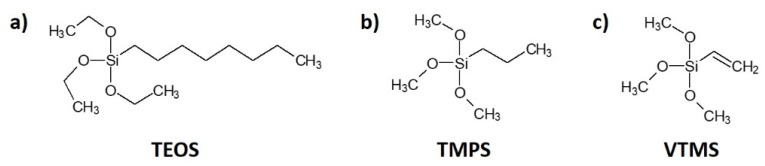
Chemical composition of the coupling agents used in the research: (**a**) triethoxyoctylsilane (TEOS); (**b**) trimethoxypropylsilane (TMPS); (**c**) vinyltrimethoxysilane (VTMS).

**Figure 2 materials-13-02901-f002:**
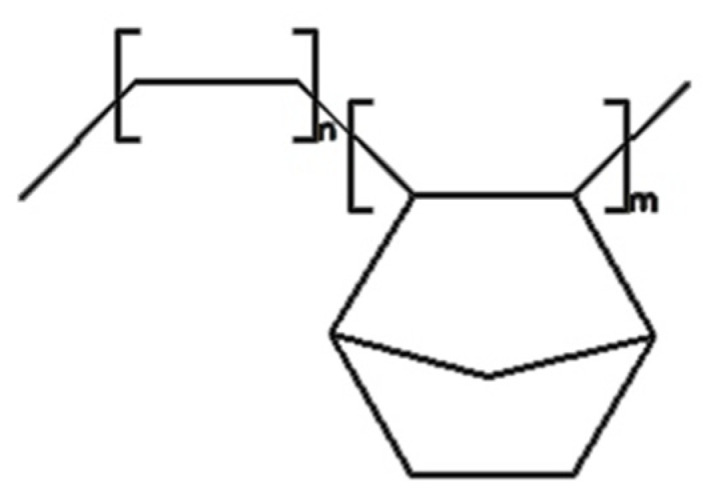
Chemical structure of the ethylene-norbornene copolymer.

**Figure 3 materials-13-02901-f003:**
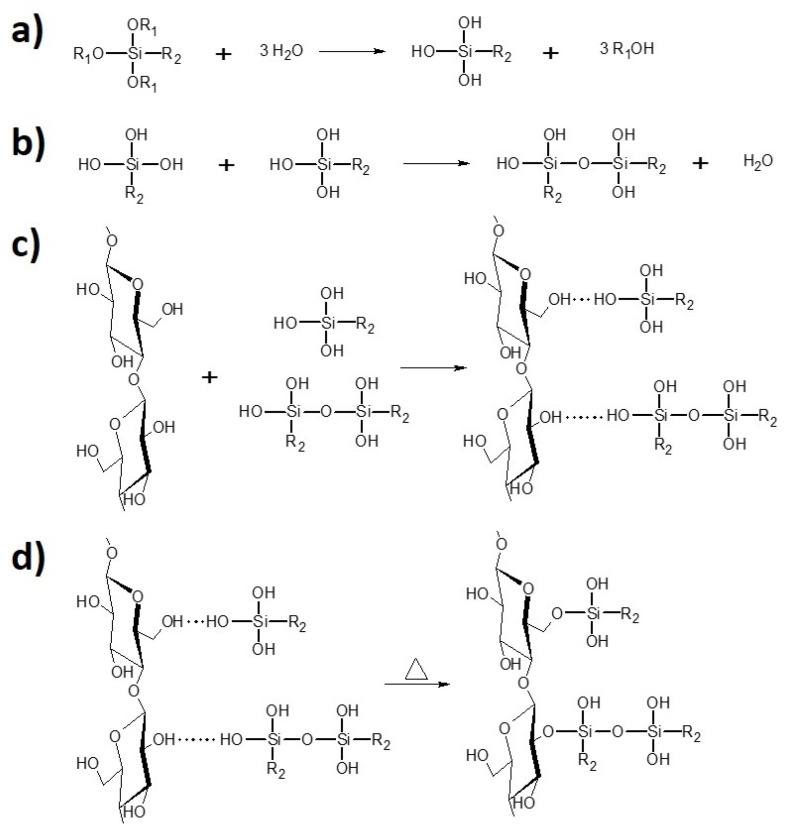
Scheme of the cellulose fibre modification with silane coupling agent: (**a**) hydrolysis; (**b**) condensation; (**c**) physical adsorption; (**d**) and chemical grafting [47].

**Figure 4 materials-13-02901-f004:**
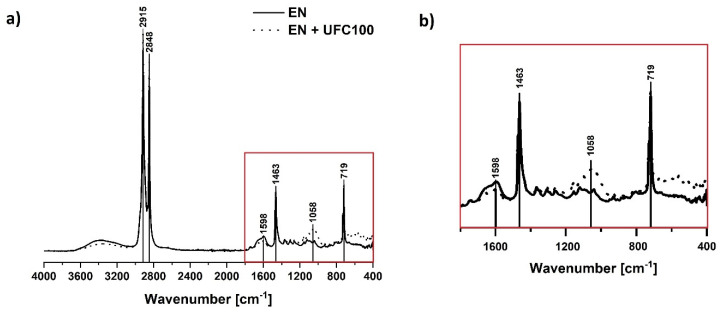
FT-IR spectra of the neat ethylene-norbornene copolymer and a composite sample filled with untreated cellulose fibers (4000–400 cm^−1^) (**a**) and its magnification (1800–400 cm^−1^) (**b**). Characteristic absorption bands: C–H stretching (2915 cm^−1^, 2848 cm^−1^), C-H bending vibrations (1463 cm^−1^), C–O, C=O, C=C, –C–O–C– bonds (1300–1100 cm^−1^), CH_2_ rocking (719 cm^−1^).

**Figure 5 materials-13-02901-f005:**
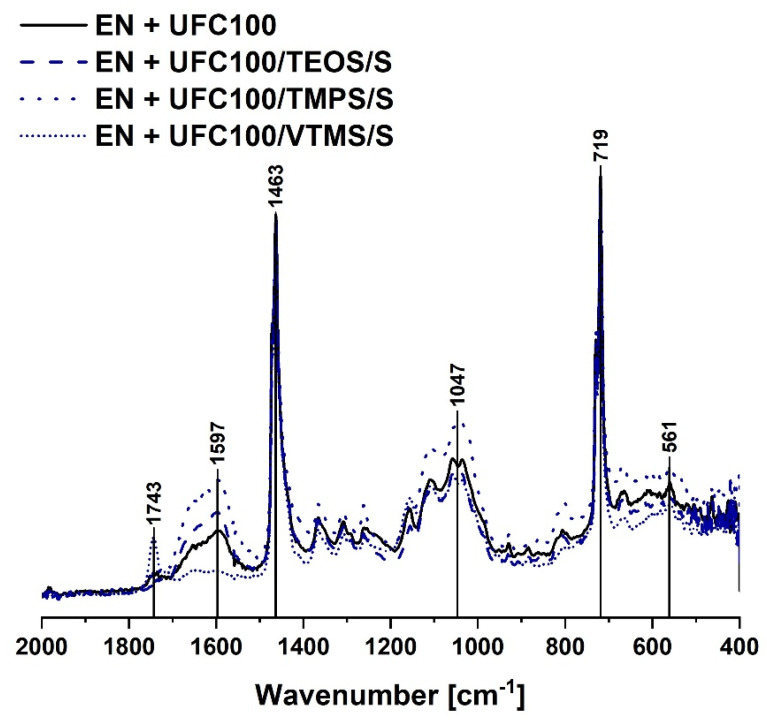
FT-IR spectra of the composite samples filled with cellulose fibers modified via a solvent-involving approach (2000–400 cm^−1^). Characteristic absorption bands: C=C bonds (1743 cm^−1^), C–H bending vibrations (1463 cm^−1^), C–O, C=O, C=C, –C–O–C– bonds (1300–1100 cm^−1^), CH_2_ rocking mode (719 cm^−1^).

**Figure 6 materials-13-02901-f006:**
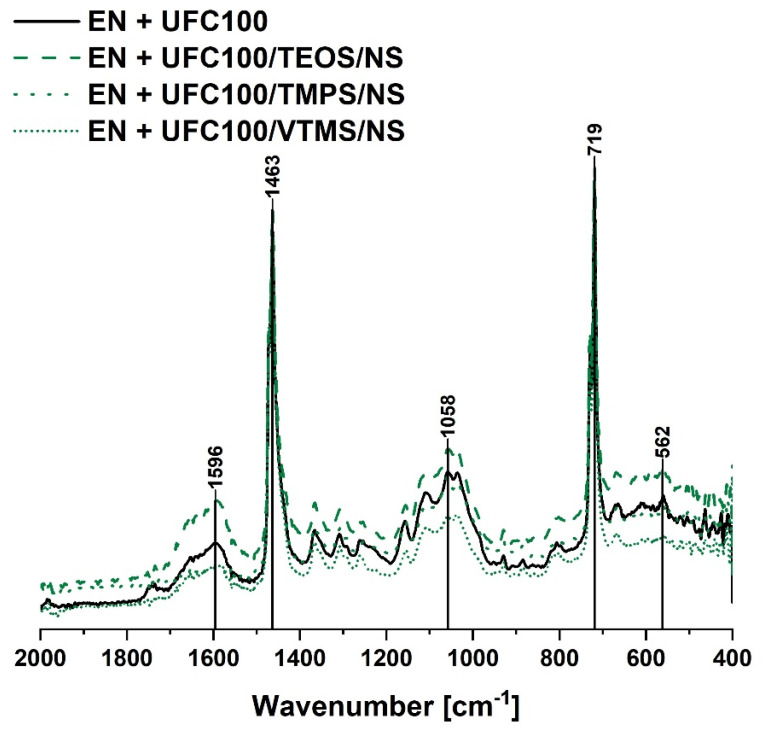
FT-IR spectra of the composite samples filled with cellulose fibers modified via a non-solvent approach (2000–400 cm^−1^). Characteristic absorption bands: C–H bending vibrations (1463 cm^−1^), C–O, C=O, C=C, –C–O–C– bonds (1300–1100 cm^−1^), CH_2_ rocking mode (719 cm^−1^).

**Figure 7 materials-13-02901-f007:**
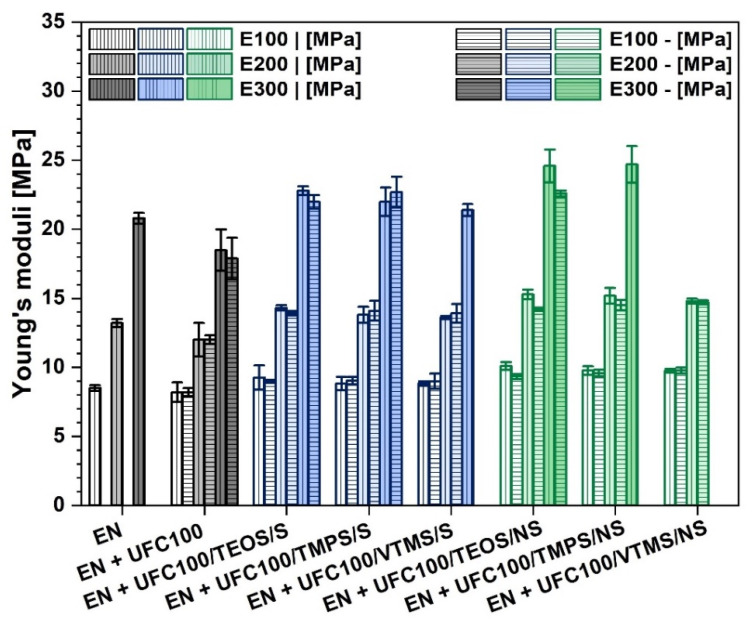
Young’s moduli of investigated composite samples: at elongation equal 100%, 200% and 300%. Samples cut out: vertically (|) and horizontally (-).

**Figure 8 materials-13-02901-f008:**
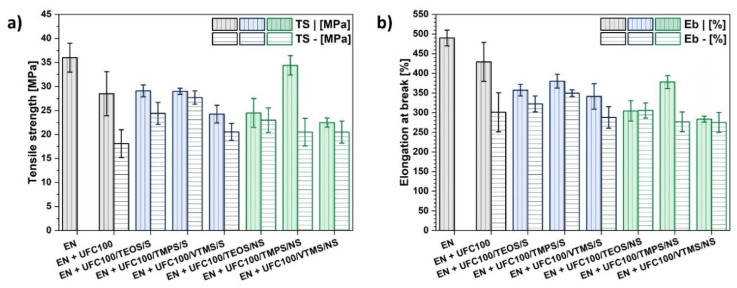
Mechanical properties of investigated composite samples: (**a**) tensile strength, (**b**) elongation at break. Samples cut out: vertically (|) and horizontally (-).

**Figure 9 materials-13-02901-f009:**
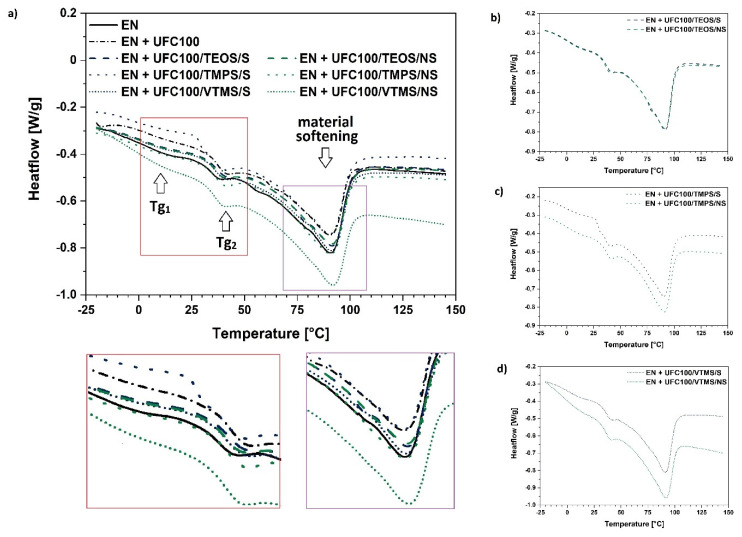
DSC curves of (**a**) all analyzed composite samples and filled with: (**b**) cellulose specimens modified with TEOS; (**c**) cellulose specimens modified with TMPS; (**d**) cellulose specimens modified with VTMS. Abbreviations: Tg_1_—glass transition of ethylene segments, Tg_2_—glass transition of norbornene rings segments.

**Figure 10 materials-13-02901-f010:**
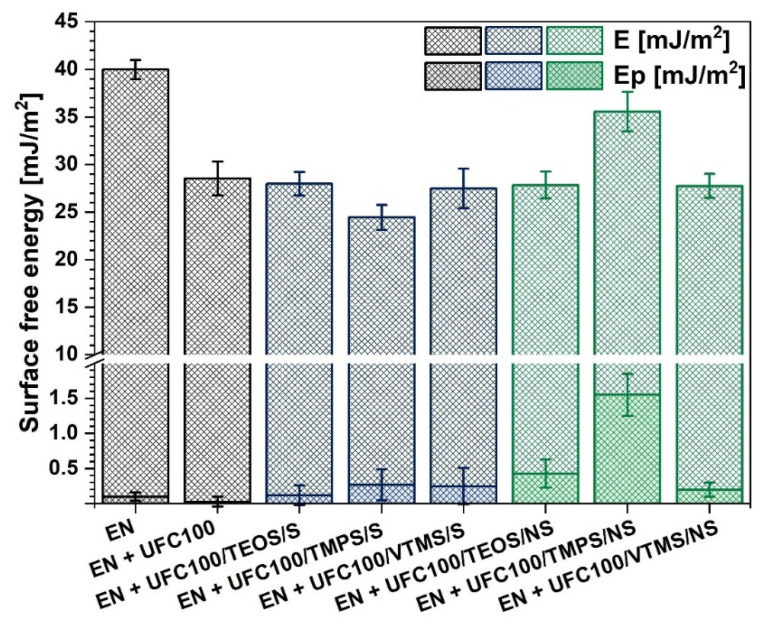
Surface free energy of investigated composites filled with modified cellulose fibers: *E*—surface free energy; *E_p_*—polar part of surface free energy.

**Figure 11 materials-13-02901-f011:**
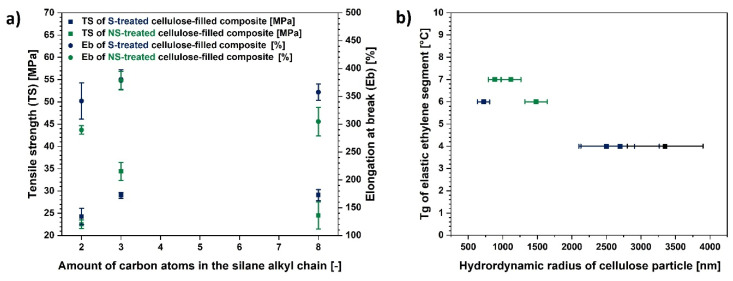
Graphs illustrating influence of: (**a**) amount of carbon atoms in the silane alkyl chain on the mechanical properties of polymer composite samples; (**b**) hydrodynamic radii of cellulose particles on the glass transition temperature of elastic ethylene segment.

**Table 1 materials-13-02901-t001:** Summary of performed cellulose modifications.

Sample	Silane			Modification Type	
	TEOS	TMPS	VTMS	Solvent-Involving Approach (S)	Non-Solvent Approach (NS)
UFC100/TEOS/S	✔	-----	-----	✔	-----
UFC100/TMPS/S	-----	✔	-----	✔	-----
UFC100/VTMS/S	-----	-----	✔	✔	-----
UFC100/TEOS/NS	✔	-----	-----	-----	✔
UFC100/TMPS/NS	-----	✔	-----	-----	✔
UFC100/VTMS/NS	-----	-----	✔	-----	✔

The effect of performed modification processes on the properties of cellulose fibers has been discussed in the *Part I* of this research [50].

**Table 2 materials-13-02901-t002:** Tabularized absorption bands assigned to the chemical groups in cellulose-filled composite samples.

Wavenumber [cm^−1^]	Chemical Group	Ref.
719	CH_2_ rocking vibrations in –(CH_2_)_n_–	[54]
1100–1000	CO-O-CO	[55]
1050	C–O stretching vibration	[54]
1100	–OH, Si–O-Si	[21,56]
1160	C–O stretching vibration	[54]
1300–1100	C–O, C=O, C=C, COOH	[56]
1460	C–H deformation vibrations in – (CH_2_)_n_–	[54]
1700–1500	C=C	[53,57]
2850	C–H symmetric stretching vibration in –CH_2_–	[52,53]
2915	C–H asymmetric stretching vibration in –CH_2_–	[53,54]
3600–3200	–OH stretching vibrations	[53,58]

**Table 3 materials-13-02901-t003:** Tabularized values of ethylene segments glass transition temperatures (Tg_1_), norbornene segments glass transition temperatures (Tg_2_), peak temperatures (T_peak_) of the softening process, and their enthalpy change (ΔH).

Sample	Tg_1_ [°C]	Tg_2_ [°C]	ΔH [J/g]	T_peak_ [°C]
EN	9	36	53,76	90
EN + UFC100	4	38	40,09	90
EN + UFC100/TEOS/S	4	38	46,26	91
EN + UFC100/TMPS/S	4	38	46,47	90
EN + UFC100/VTMS/S	6	37	47,93	91
EN + UFC100/TEOS/NS	6	36	46,24	91
EN + UFC100/TMPS/NS	7	38	46,85	91
EN + UFC100/VTMS/NS	7	37	45,24	92

**Table 4 materials-13-02901-t004:** Comparison of the polymer composite properties with LDPE.

Property	LDPE	EN + UFC100/TMPS/NS
tensile strength [MPa]	10–20 [74,75]	34 ± 2
elongation at break [%]	100–600 [74,75]	380 ± 20
degradation potential	low	higher

**Table 5 materials-13-02901-t005:** Comparison of the polymer composite properties.

Analysis	Solvent-Involving Cellulose Modification (S)	Non-Solvent Cellulose Modification (NS)
FT-IR	chemical structure of a composite unaffected; possible filler-polymer matrix interaction changes	chemical structure of a composite unaffected; possible changes in filler-polymer matrix interactions
static mechanical tests	material of a performance slightly higher than untreated cellulose-filled polymer composite	material becomes stiffer; the highest improvement in mechanical properties
DSC	effect similar to the neat UFC100 incorporation	elastic ethylene segment Tg slight increase
surface free energy	slight increase of surface free energy polar part	higher changes in polar part of surface free energy; easier wetting

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
