# Peer review of "Superiority of Cellulose Non-Solvent Chemical Modification over Solvent-Involving Treatment: Application in Polymer Composite (part II)"

_materials, 2020, doi:10.3390/ma13132901_

Round 1
Reviewer 1 Report
The paper describes a novel green approach for cellulose modification (sylanization) plus the possible application pathway for the new material as a bio-filler for polymeric composites. The materials obtained bear a great potential for their further utilisation, therefore the manuscript has a certain scientific importance; however, it requires revision before its acceptance.
The current state-of-the-art on cellulose silanysation approaches is missing. There are some studies on that which can be addresed in the introduction. Also the role of the silane coupling agents, that are used to improve the interactions between the polymer matrix and the filler, has to be mentioned. Otherwise the introduction does not represent a clear story.
Apart from that, there is no clear information on the efficiency of the modification procedure. The DS should be measured and compared for both modification approaches. It was shown that celluloses mofified by two different approaches affect the final properties of the composite in a different way. The knowledge of the DS might help to understand those differences.
Author Response
Institute of Polymer and Dye Technology
Technical University of Lodz
90-924 Lodz, ul Stefanowskiego 12/16, Poland
Tel.: +48 42 631 32 23, Fax: +48 42 636 25 43
June 15, 2020
Materials
Dear Professor,
We are resubmitting our revised paper entitled Superiority of cellulose non-solvent chemical modification over solvent-involving treatment: application in polymer composites (part II)
by Stefan Cichosz, Anna Masek with a request to reconsider it for publication in Materials.
We have carefully considered the Editor and Reviewers' comments. The manuscript
was revised exactly according to these comments. The list of responses to the reviewer’s comments and corrections made in the manuscript is attached.
The manuscript has not been previously published, is not currently submitted for review
to any other journal, and will not be submitted elsewhere before a decision is made
by this journal.
For correspondence please use the following information:
corresponding author: Anna Masek
Institute of Polymer and Dye Technology
Technical University of Lodz
90-924 Lodz, ul Stefanowskiego 12/16, Poland
Tel.: +48 42 631 32 93
Fax: +48 42 636 25 43
e-mail: [email protected]
Yours sincerely,
Ph. D., D.Sc. Anna Masek
All changes are marked with a green colour through whole manuscript.
Reviewer #1
The paper describes a novel green approach for cellulose modification (sylanization) plus the possible application pathway for the new material as a bio-filler for polymeric composites. The materials obtained bear a great potential for their further utilisation, therefore the manuscript has a certain scientific importance; however, it requires revision before its acceptance.
The comments are listed below:
- The current state-of-the-art on cellulose silanisation approaches is missing. There are some studies on that which can be addressed in the introduction. Also the role of the silane coupling agents, that are used to improve the interactions between the polymer matrix and the filler, has to be mentioned. Otherwise the introduction does not represent a clear story.
Answer: We are grateful for drawing our attention to this problem. However, we believe that the silanisation process is well discussed as it is the main topic of two paragraphs in the Introduction section – the works of Ifuku et a. and Qian et al. are described and the effect of the silane coupling agent addressed. Yet, we understand that the role of silane treatment was not stressed enough. Therefore, the Introduction has been improved and slightly changed.
- Apart from that, there is no clear information on the efficiency of the modification procedure. The DS should be measured and compared for both modification approaches. It was shown that celluloses modified by two different approaches affect the final properties of the composite in a different way. The knowledge of the DS might help to understand those differences.
Answer: We could not agree more with this comment and we are thankful for this advice as it may contribute to the improved presentation of the results. The article being a subject of this review is a second part of the bigger project and it presents the properties of the composite samples. We had not managed to define the DS before the lockdown was announced. Therefore, we are now starting to repeat the performed modifications and lacking results are going to be published in another part of the reviewed paper.
Reviewer 2 Report
I think that the manuscript entitled: “Superiority of cellulose non-solvent chemical modification over solvent-involving treatment: 3 application in polymer composite (part II)” deserves to be published in Materials under some revisions.
The aim of the paper is to prepare cellulose composites which pave new possibilities for natural fiber applications in polymer composites. Ethylene-norbornene copolymer (TOPAS) has been filled with different chemically modified cellulose fibers (UFC100). The modification of cellulose fibers has been done by using two new approaches, one of them avoiding the use of solvents which is very interesting for eco-friendly production.
However some points should be revised:
I don't understand the meaning of Part II in the title of the article. The authors do not refer to Part I even in the manuscript introduction. It would be correct to explain it. If other similar compounds have already been published by the same authors, the properties of those compounds should be briefly compared with those shown by the compounds described in the present manuscript.
On the other hand, authors do not comment anything about thermal stability. Thermogravimetric analysis is necessary to assess the thermal decomposition of investigated polymer blends. Only in line 318 it is pointed out but not explained (line 318. in most cases the degradation of the neat polymer matrix mechanical performance may be observed).
It is expected a decrease in the thermal stability of the copolymer due to the incorporation of cellulose fibers. Indeed the different modifications with TEOS, TMPS and VTMS should affect to the degree of crystallinity of cellulose, a factor responsible for thermal stability.
Line 164. Please, better change in the figure legend 1), 2), 3) and 4) by a), b), c) and d).
Figure 4. Please, label the IR spectrum with the same values indicated in the text (2447 instead 2448 and 1460 instead 1463). In Table 2 is 719 instead 720?. Please, clarify if 719 is for EN and 720 is for EN-UFC 100. Are the bands labelled in figure 4 those belonging to neat polymer or to cellulose- filled polymer?
Why the bands in the EN+UFC 100 have lower intensity? Have been the infrared spectra normalized?
Paragraph between lines 263 and 268 should be before figure 5
Figure 9a. It is really difficult to observe the differences in the graphs. It would be better to use different colours in order to better differentiate between modified composites with solvent and non-solvent approach
Fig 11. There is a typing mistake (composie)
Table 4, before at the end of results and discussion part. Comparison of cellulose-filled composite properties with those of LDPE must be in the discussion part.
Conclusion part must be improved. In general, there is an uncertainty on the part of the authors in the conclusions: ” analysed material may degrade more easily” “Copolymer composite may become an interesting alternative” What does may mean? It has to be clear and make sure that the new composites do degrade and represent a better alternative to the materials that currently exist on the market.
Author Response
Institute of Polymer and Dye Technology
Technical University of Lodz
90-924 Lodz, ul Stefanowskiego 12/16, Poland
Tel.: +48 42 631 32 23, Fax: +48 42 636 25 43
June 15, 2020
Materials
Dear Professor,
We are resubmitting our revised paper entitled Superiority of cellulose non-solvent chemical modification over solvent-involving treatment: application in polymer composites (part II)
by Stefan Cichosz, Anna Masek with a request to reconsider it for publication in Materials.
We have carefully considered the Editor and Reviewers' comments. The manuscript
was revised exactly according to these comments. The list of responses to the reviewer’s comments and corrections made in the manuscript is attached.
The manuscript has not been previously published, is not currently submitted for review
to any other journal, and will not be submitted elsewhere before a decision is made
by this journal.
For correspondence please use the following information:
corresponding author: Anna Masek
Institute of Polymer and Dye Technology
Technical University of Lodz
90-924 Lodz, ul Stefanowskiego 12/16, Poland
Tel.: +48 42 631 32 93
Fax: +48 42 636 25 43
e-mail: [email protected]
Yours sincerely,
Ph. D., D.Sc. Anna Masek
All changes are marked with a green colour through whole manuscript.
Reviewer #2
I think that the manuscript entitled: “Superiority of cellulose non-solvent chemical modification over solvent-involving treatment: application in polymer composite (part II)” deserves to be published in Materials under some revisions. The aim of the paper is to prepare cellulose composites which pave new possibilities for natural fiber applications in polymer composites. Ethylene-norbornene copolymer (TOPAS) has been filled with different chemically modified cellulose fibers (UFC100). The modification of cellulose fibers has been done by using two new approaches, one of them avoiding the use of solvents which is very interesting for eco-friendly production.
The comments are listed below:
- I don't understand the meaning of Part II in the title of the article. The authors do not refer to Part I even in the manuscript introduction. It would be correct to explain it. If other similar compounds have already been published by the same authors, the properties of those compounds should be briefly compared with those shown by the compounds described in the present manuscript.
Answer: We agree that this title could be misleading. During the revision of Part II of this article, also Part I has been under revision. The information about Part I has been incorporated in 2.2. Modification of cellulose fibres section: The effect of performed modification processes on the properties of cellulose fibres has been discussed in the Part I of this research [50].
- On the other hand, authors do not comment anything about thermal stability. Thermogravimetric analysis is necessary to assess the thermal decomposition of investigated polymer blends. Only in line 318 it is pointed out but not explained (line 318. in most cases the degradation of the neat polymer matrix mechanical performance may be observed).
Answer: The sentence in line 318 is connected with the degradation of mechanical properties and not the thermal ones. The whole paragraph: Furthermore, Fig. 8 reveals the tensile strength and elongation at break values for analysed polymer composite samples. In most cases the degradation of the neat polymer matrix mechanical performance may be observed. The reinforcing effect of the modified cellulose only slightly differs from untreated natural fibres. In the next paragraph an introduced idea is explained: Nevertheless, reproducing the tensile strength of the neat polymer matrix is possible. Regarding Fig. 8a, cellulose modified with TMPS via a non-solvent approach, while incorporated into the ethylene-norbornene copolymer, significantly reinforce the material creating a product of a similar performance to the neat polymer matrix.
Nevertheless, we understand that the word degradation in this paragraph could be misleading. Therefore, the sentence has been rewritten as follows: In most cases the lowering of composite mechanical performance may be observed.
Considering the lacking information about the thermal stability of the developed polymer composites, we would like to underline that due to the lockdown, the research has prolonged and, therefore, we are going to publish more results of further investigation in the future (another part of the research).
- It is expected a decrease in the thermal stability of the copolymer due to the incorporation of cellulose fibers. Indeed the different modifications with TEOS, TMPS and VTMS should affect to the degree of crystallinity of cellulose, a factor responsible for thermal stability.
Answer: We are thankful for this advice. The article being a subject of this review is a second part of the bigger project and it presents the properties of the composite samples. We had not managed to define the crystallinity degree before the lockdown was announced. Therefore, we are now starting to repeat the performed modifications and lacking results are going to be published in another part of the reviewed paper.
- Line 164. Please, better change in the figure legend 1), 2), 3) and 4) by a), b), c) and d).
Answer: We are grateful for this comment. The legend was corrected.
- Figure 4. Please, label the IR spectrum with the same values indicated in the text (2447 instead 2448 and 1460 instead 1463). In Table 2 is 719 instead 720?. Please, clarify if 719 is for EN and 720 is for EN-UFC 100. Are the bands labelled in figure 4 those belonging to neat polymer or to cellulose- filled polymer?
Answer: We have corrected the attributed wavenumbers. The title of Table 2 has been corrected: Table 2 Tabularized absorption bands assigned to the chemical groups in cellulose-filled composite samples.
- Why the bands in the EN+UFC 100 have lower intensity? Have been the infrared spectra normalized?
Answer: Yes, the spectra was normalized. It is not the whole spectra which has a lower intensity, but only some specific regions which are attributed to the particular chemical groups. It is expected that the filled system interacts differently with the IR irradiation in comparison with the neat polymer matrix. Differences could be also caused by the water content in cellulose fibres. The description of the FT-IR results has been improved and revised.
- Paragraph between lines 263 and 268 should be before figure 5.
Answer: We are thankful for drawing our attention to this problem. The mistake has been corrected.
- Figure 9a. It is really difficult to observe the differences in the graphs. It would be better to use different colours in order to better differentiate between modified composites with solvent and non-solvent approach.
Answer: We would insist on leaving the colours as they are chosen in order to stay constant through the whole article. Yet, we have improved the graphic.
- Fig 11. There is a typing mistake (composie).
Answer: We are sorry for this typo. The mistake is corrected.
- Table 4, before at the end of results and discussion part. Comparison of cellulose-filled composite properties with those of LDPE must be in the discussion part.
Answer: We are thankful for this advice. The new section has been created:
3.6. Comparison of prepared composite properties with LDPE.
- Conclusion part must be improved. In general, there is an uncertainty on the part of the authors in the conclusions: ” analysed material may degrade more easily” “Copolymer composite may become an interesting alternative” What does may mean? It has to be clear and make sure that the new composites do degrade and represent a better alternative to the materials that currently exist on the market.
Answer: We meant to introduce an idea and not the statement. Nevertheless, we understand that this phrase could be misleading. The Conclusions section has been improved.
Reviewer 3 Report
The authors did good research for the manuscript. There are some suggestions for authors:
line 52-63, the authors can put more references into content to prove it.
line 77-89, the authors can put more references into content to prove it.
in the conclusions, the Table 4 can move out from the paper, just use words to explain conclusions.
Author Response
Institute of Polymer and Dye Technology
Technical University of Lodz
90-924 Lodz, ul Stefanowskiego 12/16, Poland
Tel.: +48 42 631 32 23, Fax: +48 42 636 25 43
June 15, 2020
Materials
Dear Professor,
We are resubmitting our revised paper entitled Superiority of cellulose non-solvent chemical modification over solvent-involving treatment: application in polymer composites (part II)
by Stefan Cichosz, Anna Masek with a request to reconsider it for publication in Materials.
We have carefully considered the Editor and Reviewers' comments. The manuscript
was revised exactly according to these comments. The list of responses to the reviewer’s comments and corrections made in the manuscript is attached.
The manuscript has not been previously published, is not currently submitted for review
to any other journal, and will not be submitted elsewhere before a decision is made
by this journal.
For correspondence please use the following information:
corresponding author: Anna Masek
Institute of Polymer and Dye Technology
Technical University of Lodz
90-924 Lodz, ul Stefanowskiego 12/16, Poland
Tel.: +48 42 631 32 93
Fax: +48 42 636 25 43
e-mail: [email protected]
Yours sincerely,
Ph. D., D.Sc. Anna Masek
All changes are marked with a green colour through whole manuscript.
Reviewer #3
The authors did good research for the manuscript.
The comments are listed below:
- Line 52-63 and Line 77-89, the authors can put more references into content
to prove it.
Answer: These are the results of the research done by Shumigin et al. Further, in the next paragraphs, the presented results are compared with different research studies. Nevertheless, we understand that this may be misleading. Therefore some new references has been added as requested.
- In the conclusions, the Table 4 can move out from the paper, just use words to explain conclusions.
Answer: We are thankful for this comment. However, we believe that comparison presented in the table is easier to read than the comparison within the lines of the text. Therefore, we would like to leave the Table 4 (new: Table 5) in the Conclusions section. Yet, this part has been revised and improved.
Reviewer 4 Report
This paper deals with the preparation of silane-treated cellulose fibres and their use for the preparation of ethylene-norbonene copolymer/cellulose composites. Cellulose fibres modified with silane derivatives were prepared by two different methods: with solvent and without solvent.
The topic of the paper is interesting considering the possibility to improve the properties of EN copolymer by using a bio-filler. However, in my opinion, the paper in this form lacks of information. For example, the characterization of the modified fibres is missing and no data are reported about the grafting of silane groups (qualitatively and quantitatively) as well as about fibre dimensions. About this last point, in Fig. 11b it is reported the hydrodynamic radius of the cellulose fibres, but no indication of its determination is given neither in experimental part nor in results and discussion. Even if from the title of the paper I understand that it is the second part of a research study, in my opinion the characterization of the fibres should be reported. For example, in order to better compare the two methods of modification of cellulose fibres, data about the chemical grafting of silanes should be added. It could be interesting to know if the modification degree for the same type of silane depends on the type of modification process or not and how this characteristic may influence the dispersion into the polymer matrix as well as the polymer-filler interaction. Moreover, the manuscript is incremental with respect to previous papers of the authors dealing with the modification of cellulose with VTMS (S. Cichosz et al., Polymer Bulletin 76 (2019) 2147) and with the preparation of EN copolymer/VTMS-cellulose composites (S. Cichosz et al. Polym. Degrad. Stab. 159 (2019) 174).
Please, consider also the following major and minor points.
Section 2.3 “Preparation of polymer composite samples”. Please, check the quantity of polymer and filler: is the cellulose quantity 7 wt% or 14 wt%?
Pag. 5: the scheme reported in Fig. 3 is for a generic silane not only for VTMS. Moreover, in the scheme, reactions are labeled with letters and in the caption with numbers.
Section 2.4: the title is “Characterization of cellulose fibres”, but the characterizations reported are about polymer composites. Please, check.
Section 2.4.2: in this section it is reported that in order to observe the fibre orientation effect, specimen were cut out in two directions. However it is not clear to me how it was possible to identify a vertically or horizontally direction in films obtained in a hydraulic press. Please, could you explain?
In both Fig. 4 and 5 the differences between the spectra are barely discernible. Maybe a magnification of selected regions could help.
Please, the FTIR signal at 719 cm-1 is likely due to CH2 rocking vibration of ethylene monomeric units and not C-C.
Pag. 8 lines 276-278. Please, check this part “Moreover, considering the composite sample filled with VTMS modified cellulose via a solvent-involving approach, an additional peak at 1743 cm-1 (C=C, C=O) [47,51] is visible. This may indicate some possible bonding between the polymer matrix and a silane coupling agent grafted on the surface of the fibres”. It is not clear to me how these signals can be correlated to a bond between EN copolymer and VTMS modified cellulose. Please, could you explain.
The results of static mechanical analysis depend on dispersion and distribution of the fibres into the polymer matrix. Accordingly, a morphological characterization of polymer composites could help in understanding the results. Moreover, also a different degree of modification of the fibres could alter their dispersion and interaction with the matrix. These two points should be taken into consideration in the paper to explain the results.
Please, check the caption of Fig. 7: a, b and c are not reported in the picture.
Pag. 12 lines 361-363. The authors wrote: “Generally, Tg1 values are higher in case of cellulose fibres modified via a mechano-chemical approach and treatments performed with VTMS”. However, data reported in tab. 3 are similar also for other composites.
Author Response
Institute of Polymer and Dye Technology
Technical University of Lodz
90-924 Lodz, ul Stefanowskiego 12/16, Poland
Tel.: +48 42 631 32 23, Fax: +48 42 636 25 43
June 15, 2020
Materials
Dear Professor,
We are resubmitting our revised paper entitled Superiority of cellulose non-solvent chemical modification over solvent-involving treatment: application in polymer composites (part II)
by Stefan Cichosz, Anna Masek with a request to reconsider it for publication in Materials.
We have carefully considered the Editor and Reviewers' comments. The manuscript
was revised exactly according to these comments. The list of responses to the reviewer’s comments and corrections made in the manuscript is attached.
The manuscript has not been previously published, is not currently submitted for review
to any other journal, and will not be submitted elsewhere before a decision is made
by this journal.
For correspondence please use the following information:
corresponding author: Anna Masek
Institute of Polymer and Dye Technology
Technical University of Lodz
90-924 Lodz, ul Stefanowskiego 12/16, Poland
Tel.: +48 42 631 32 93
Fax: +48 42 636 25 43
e-mail: [email protected]
Yours sincerely,
Ph. D., D.Sc. Anna Masek
All changes are marked with a green colour through whole manuscript.
Reviewer #4
This paper deals with the preparation of silane-treated cellulose fibres and their use for the preparation of ethylene-norbornene copolymer/cellulose composites. Cellulose fibres modified with silane derivatives were prepared by two different methods: with solvent and without solvent.
The topic of the paper is interesting considering the possibility to improve the properties of EN copolymer by using a bio-filler. However, in my opinion, the paper in this form lacks of information. For example, the characterization of the modified fibres is missing and no data are reported about the grafting of silane groups (qualitatively and quantitatively) as well as about fibre dimensions. About this last point, in Fig. 11b it is reported the hydrodynamic radius of the cellulose fibres, but no indication of its determination is given neither in experimental part nor in results and discussion. Even if from the title of the paper I understand that it is the second part of a research study, in my opinion the characterization of the fibres should be reported. For example, in order to better compare the two methods of modification of cellulose fibres, data about the chemical grafting of silanes should be added. It could be interesting to know if the modification degree for the same type of silane depends on the type of modification process or not and how this characteristic may influence the dispersion into the polymer matrix as well as the polymer-filler interaction. Moreover, the manuscript is incremental with respect to previous papers of the authors dealing with the modification of cellulose with VTMS (S. Cichosz et al., Polymer Bulletin 76 (2019) 2147) and with the preparation of EN copolymer/VTMS-cellulose composites (S. Cichosz et al. Polym. Degrad. Stab. 159 (2019) 174).
General answer: We are grateful for this broad comment. It was very valuable.
The information about the previous part of this article was introduced into the given manuscript and the information about the determination of cellulose sample hydrodynamic radii is added. Moreover, we would like to add that not all information gathered during this broad research are published and we are working on the next parts clarifying the insecurities.
The comments are listed below:
- Section 2.3 “Preparation of polymer composite samples”. Please, check the quantity of polymer and filler: is the cellulose quantity 7 wt% or 14 wt%?
Answer: We are sorry for this mistake. It is corrected.
- Pag. 5: the scheme reported in Fig. 3 is for a generic silane not only for VTMS. Moreover, in the scheme, reactions are labelled with letters and in the caption with numbers.
Answer: We are thankful for drawing our attention to this problem – we have corrected the caption.
- Section 2.4: the title is “Characterization of cellulose fibres”, but the characterizations reported are about polymer composites. Please, check. Section 2.4.2: in this section it is reported that in order to observe the fibre orientation effect, specimen were cut out in two directions. However it is not clear to me how it was possible to identify a vertically or horizontally direction in films obtained in a hydraulic press. Please, could you explain?
Answer: Of course, we have meant the Characterization of cellulose-filled composites. Mistake is corrected. Considering the problem of the fibres orientation within the polymer matrix, we try to align the fibres with two rolling mills before the preparation of composite plates in the hydraulic press. Then we cut out the samples in two directions: vertically and horizontally. Next, while the tensile testing, it is possible to observe that the elongation at break and tensile strength values are higher in one of the directions (the direction of rolling in two rolling mills), which we assume, is a possible direction of the cellulose fibres orientation within the polymer matrix. The procedure is described in 2.3. Preparation of polymer composite samples section.
- In both Fig. 4 and 5 the differences between the spectra are barely discernible. Maybe a magnification of selected regions could help.
Answer: We are thankful for this comment. The figures have been improved considering the size and readability.
- Please, the FTIR signal at 719 cm-1 is likely due to CH2 rocking vibration of ethylene monomeric units and not C-C.
Answer: We are grateful for this comment. Mistake is corrected.
- Pag. 8 lines 276-278. Please, check this part “Moreover, considering the composite sample filled with VTMS modified cellulose via a solvent-involving approach, an additional peak at 1743 cm-1 (C=C, C=O) [47,51] is visible. This may indicate some possible bonding between the polymer matrix and a silane coupling agent grafted on the surface of the fibres”. It is not clear to me how these signals can be correlated to a bond between EN copolymer and VTMS modified cellulose. Please, could you explain.
Answer: It was only an introduction of the possible idea. Yet, we understand that this may be misleading. Therefore this sentence has been removed.
- The results of static mechanical analysis depend on dispersion and distribution of the fibres into the polymer matrix. Accordingly, a morphological characterization of polymer composites could help in understanding the results. Moreover, also a different degree of modification of the fibres could alter their dispersion and interaction with the matrix. These two points should be taken into consideration in the paper to explain the results.
Answer: We are grateful for this comment and we thank for drawing our attention to this problem. We agree that these information would be very helpful. Yet, due to the lockdown, we are unable to carry out all the experiments. Moreover, we would like to add that not all information gathered during this broad research are published and we are working on the next parts clarifying the insecurities. In future, we plan to develop this research.
Nevertheless, to stay true and reliable, we have addressed the issues mentioned by the Reviewer: This could be explained by the fact that during such a treatment, not only the cellulose fibres grafting with silane coupling agent occurs, but also the particle size decreases due to the milling process [59,60]. The combination of an efficient hydrophobisation of natural fibres and appropriate particle size distribution may be the way to obtain a cellulose-reinforced polymer composite. Nevertheless, changes in the filler dispersion and composite morphology should be taken into consideration in order to fully understand the ongoing variations in mechanical properties.
- Please, check the caption of Fig. 7: a, b and c are not reported in the picture.
Answer: Yes. We agree that this could have been misleading. Therefore, the caption was corrected.
- Pag. 12 lines 361-363. The authors wrote: “Generally, Tg1 values are higher in case of cellulose fibres modified via a mechano-chemical approach and treatments performed with VTMS”. However, data reported in tab. 3 are similar also for other composites.
Answer: We agree that this statement is uncertain. Therefore it was removed.
Round 2
Reviewer 2 Report
The new revised-draft of the manuscript is correct and deserves to be published in Materials
Author Response
Institute of Polymer and Dye Technology
Technical University of Lodz
90-924 Lodz, ul Stefanowskiego 12/16, Poland
Tel.: +48 42 631 32 23, Fax: +48 42 636 25 43
June 19, 2020
Materials
Dear Professor,
We are resubmitting our revised paper entitled Superiority of cellulose non-solvent chemical modification over solvent-involving treatment: application in polymer composites (part II)
by Stefan Cichosz, Anna Masek with a request to reconsider it for publication in Materials.
We have carefully considered the Editor and Reviewers' comments. The manuscript
was revised exactly according to these comments. The list of responses to the reviewer’s comments and corrections made in the manuscript is attached.
The manuscript has not been previously published, is not currently submitted for review
to any other journal, and will not be submitted elsewhere before a decision is made
by this journal.
For correspondence please use the following information:
corresponding author: Anna Masek
Institute of Polymer and Dye Technology
Technical University of Lodz
90-924 Lodz, ul Stefanowskiego 12/16, Poland
Tel.: +48 42 631 32 93
Fax: +48 42 636 25 43
e-mail: [email protected]
Yours sincerely,
Ph. D., D.Sc. Anna Masek
All changes are marked with a green colour through whole manuscript.
Reviewer #2
The new revised-draft of the manuscript is correct and deserves to be published in Materials
Reviewer 4 Report
The revised manuscript has improved taking into account the comments. However, I still have some issues. Please, before publication these points should be addressed:
- pag. 8, lines 276-278. As I mentioned in my previous report, I have some doubts about this point. The peak at 1743 cm-1 can not be assigned to silane C=C bond, because the C=C silane stretching is at about 1600 cm-1. Please, check in the literature.
- pag. 7, lines 243. In this part of the text it is still C-C rocking mode instead of CH2
- In the reference 50, it lacks the journal. Is this paper submitted? is it under review? Please, add information.
Author Response
Institute of Polymer and Dye Technology
Technical University of Lodz
90-924 Lodz, ul Stefanowskiego 12/16, Poland
Tel.: +48 42 631 32 23, Fax: +48 42 636 25 43
June 19, 2020
Materials
Dear Professor,
We are resubmitting our revised paper entitled Superiority of cellulose non-solvent chemical modification over solvent-involving treatment: application in polymer composites (part II)
by Stefan Cichosz, Anna Masek with a request to reconsider it for publication in Materials.
We have carefully considered the Editor and Reviewers' comments. The manuscript
was revised exactly according to these comments. The list of responses to the reviewer’s comments and corrections made in the manuscript is attached.
The manuscript has not been previously published, is not currently submitted for review
to any other journal, and will not be submitted elsewhere before a decision is made
by this journal.
For correspondence please use the following information:
corresponding author: Anna Masek
Institute of Polymer and Dye Technology
Technical University of Lodz
90-924 Lodz, ul Stefanowskiego 12/16, Poland
Tel.: +48 42 631 32 93
Fax: +48 42 636 25 43
e-mail: [email protected]
Yours sincerely,
Ph. D., D.Sc. Anna Masek
All changes are marked with a green colour through whole manuscript.
Reviewer #4
The revised manuscript has improved taking into account the comments. However, I still have some issues. Please, before publication these points should be addressed.
The comments are listed below:
- Pag. 8, lines 276-278. As I mentioned in my previous report, I have some doubts about this point. The peak at 1743 cm-1 cannot be assigned to silane C=C bond, because the C=C silane stretching is at about 1600 cm-1. Please, check in the literature.
Answer: We are thankful for this comment and deep analysis of the FT-IR spectra. The explanation given in the article has been improved.
- Pag. 7, lines 243. In this part of the text it is still C-C rocking mode instead of CH2
Answer: The sentence has been corrected.
- In the reference 50, it lacks the journal. Is this paper submitted? is it under review? Please, add information.
Answer: We are sorry for this mistake. We did not see that the Mendeley software skipped some data.